# Parental Monitoring and Adolescent Problematic Mobile Phone Use: The Mediating Role of Escape Motivation and the Moderating Role of Shyness

**DOI:** 10.3390/ijerph17051487

**Published:** 2020-02-25

**Authors:** Xinchen Fu, Jingxuan Liu, Ru-De Liu, Yi Ding, Jia Wang, Rui Zhen, Fangkai Jin

**Affiliations:** 1Beijing Key Laboratory of Applied Experimental Psychology, National Demonstration Center for Experimental Psychology Education (Beijing Normal University), Faculty of Psychology, Beijing Normal University, Beijing 100875, China; fxc_psy@163.com (X.F.); jfk2016@163.com (F.J.); 2Department of Psychology and Neuroscience, Duke University, Durham, NC 27708, USA; liujingxuan1503@gmail.com; 3Graduate School of Education, Fordham University, New York, NY 10023, USA; yding4@fordham.edu; 4Teachers’ College, Beijing Union University, Beijing 100011, China; wangjia@mail.bnu.edu.cn; 5Institute of Psychological Sciences, College of Education, Hangzhou Normal University, No. 2318 Yuhangtang Road, Hangzhou 311121, China; zhenrui1206@126.com

**Keywords:** parental monitoring, problematic mobile phone use, escape motivation, shyness

## Abstract

In an attempt to reduce the negative consequences of adolescent media use, parents often monitor their children’s online activities. However, research suggests that parental monitoring often does not reduce children’s problematic mobile phone use as expected. Based on the results of a survey of 584 Chinese adolescents, we found that parental monitoring positively predicted children’s problematic mobile phone use (PMPU) within a Chinese cultural context. The results also showed that children’s escape motivation partially mediated this relationship, while their level of shyness moderated both the mediated path and the direct impact of parental monitoring on children’s PMPU. The findings suggested that a higher level of shyness increased the likelihood that parental monitoring would increase the child’s escape motivation and PMPU. The study results provide guidelines for parents and educators regarding interventions for adolescents’ problematic phone use.

## 1. Introduction

Mobile phones are the most common platform used by adolescents to access the internet. In Europe, adolescents use mobile phones more frequently than they do computers to access online information [1]. In China, 98.6% of internet users, or 817 million people, of whom 18% are adolescents, connect to the internet through mobile phones [2]. Some common internet activities adolescents engage in via mobile phones are socializing [3,4], accessing entertainment [5,6], and obtaining study resources [7].

Because mobile phones offer such a wide range of attractions, some adolescents overuse their mobile phones, which can lead to problematic mobile phone use (PMPU) [8,9]. PMPU, also known as mobile phone dependency [10] and mobile phone addiction [11], can elicit unpleasant withdrawal symptoms when an individual’s mobile phone access is restricted. Studies have found that adolescents are more prone to mobile phone addiction than adults [12,13].

Considering the wide range of negative health and psychological consequences of PMPU for adolescents [14,15,16], parents often mediate their child’s mobile phone use in an attempt to reduce PMPU. One type of mediation strategy used frequently by parents, parental monitoring, has drawn wide social attention [17]. Parental monitoring refers to parents’ tracking of children’s online activities [18] and is often thought by parents to be an effective way to prevent their child’s PMPU. However, research has shown that parental monitoring may not reduce adolescent PMPU behaviors as expected [8,19,20]. Therefore, the current study aimed to further explore the impact of parental monitoring on children’s problematic mobile phone use and to investigate the mechanism underlying such an impact in the Chinese cultural context.

### 1.1. Parental Monitoring and Problematic Mobile Phone Use (PMPU)

PMPU is a type of behavioral addiction [21,22]. It can produce a series of negative health and psychological consequences, such as sleep problems and physical harm [16,23]; decreased level of physical activity [24,25]; social problems [16]; decreased life satisfaction [26]; academic problems [15,27,28,29]; anxiety when separated from mobile phones [30,31]; negative emotions including stress, anxiety, and depression [14,32]; alexithymia [33,34]; and decreased empathy [26].

In an attempt to reduce the negative effects of children’s media use, parents often adopt different mediation strategies to influence their child’s media-use behaviors [35]. Parental mediation is a multi-dimensional concept. It encompasses all types of parental strategies, including mediation, controlling, and providing instruction and interpretation regarding media content, to reduce the negative effect of media use on children [36]. Parental mediation was originally divided into three dimensions: restrictive mediation, active mediation, and co-use [37]. These three dimensions were found to be present in parental mediation of children’s TV viewing [38,39], gaming [40], and internet use behaviors [35]. However, as internet use became more prevalent among adolescents, researchers suggested that the traditional three dimensions needed to be expanded to further address parental strategies regarding children’s internet use behaviors [17,41,42]. Nikken and Jansz [18] proposed a five-dimensional model, in which the restrictive mediation dimension was divided into time restriction and special content restriction, and supervision (i.e., monitoring) was added as a new dimension. Parental monitoring refers to parents’ tracking of their child’s online activities and history, such as email interactions and website access [17,18,20].

Parental monitoring is the most common mediation strategy for adolescent internet use [43] and can predict adolescent internet-use behaviors [44]. Yet surprisingly, many studies have found that parental monitoring is ineffective in reducing those problematic behaviors. Livingstone and Helsper [17] found that parental monitoring failed to reduce risky online behaviors (such as porn, violence, and privacy viewing) in children aged 9–16 years; a follow-up study showed that parental monitoring was positively correlated with increased internet-related risk [45]. Studies also found that parental monitoring positively predicted children’s internet-use behaviors [44] and adolescent internet addiction [46]. In addition, cross-cultural research conducted by Bayraktar [47] found that whereas parental monitoring was negatively correlated with adolescent risky internet-use behaviors (such as porn, viewing violence and excessive internet use) in Europe, it positively predicted adolescent involvement in such risky behaviors in Turkey. With increased adolescent access to the internet via mobile phones, parental monitoring of internet use has evolved to monitoring of internet-related mobile phone-use behaviors [48]. A study conducted in Germany [19] found the correlation between parental monitoring and adolescent mobile phone dependency to be insignificant: Parental monitoring was inefficient in reducing the negative effects of children’s PMPU. Similar results were obtained in Taiwan [8]. These study results suggested that parental monitoring may not reduce children’s PMPU. The first goal of our study was to examine whether parental monitoring was related to or positively predicted adolescent PMPU.

### 1.2. Mediating Role of Escape Motivation

Parental monitoring refers to parental supervision of children’s internet use behaviors with the aim to reduce the negative impacts of media use [48]. Why would parental monitoring perpetuate children’s PMPU behaviors instead of reducing them? The underlying mechanism warrants further investigation. Based on the existing theories and study results, we hypothesized that escape motivation might mediate the path between parental monitoring and children’s PMPU.

Within the context of adolescent PMPU, escape motivation refers to the motivation that drives adolescents to escape negative emotions via mobile phone use, leading to PMPU [49]. Escaping reality is one of two major functions of PMPU [50]. In many cases, the underlying motivation of addictive behaviors is to escape reality in an effort to reduce painful and negative emotions [51]. Among all types of motivation, escape motivation is often thought to be the most important predictor of internet addiction; it serves as a strong predictor for a wide range of internet and mobile phone-related addictions, including internet addiction [52,53,54], video game addiction [55,56], online game addiction [57,58], online video apps overuse (i.e., YouTube) [59], and PMPU [60,61,62].

Parental monitoring can also induce negative emotions in adolescents. According to the self-determination theory, all humans strive for freedom, and therefore their motivation is optimal when they are void of external influence and interference [63]. Thus, when children’s freedom is restricted, they may experience reactance [64]. Some studies have already examined the mediating roles of escape motivation in the relations between negative emotions, addictive behaviors, and PMPU. For example, escape motivation was found to mediate the relation between psychological distress (such as emotional imbalance, depression, and anxiety) and video game addiction [65,66] and the relation between psychiatric disorders (such as somatization and OCD) and video game addiction [67]. Escape motivation was also found to mediate the path from loneliness to PMPU [67]. Accordingly, parental monitoring may result in conflicts between parents and child, and thus cause the child to experience stress and negative emotions. As a result, through escape motivation, the child’s mobile phone-use behaviors might increase [68,69], leading to PMPU [62].

### 1.3. Moderating Role of Shyness

When choosing mediation strategies to reduce children’s PMPU, parents should take their child’s temperament into consideration [70]. Shyness, one of the most stable temperament types, should be given special attention [71].

Shyness is a common social experience that involves timidity, discomfort, embarrassment, and fear of being evaluated. It is often accompanied by a desire to minimize social interactions [72,73,74]. Research suggests that shyness can induce a series of negative consequences, including loneliness [75], anxiety, and depression [76,77]. To reduce or avoid those negative consequences, shy individuals often view online communication as a means to avoid face-to-face interaction [77]. An individual’s degree of shyness can positively predict online social activities [78] and internet addictive behaviors [79,80]. Such a predictive effect was shown to be consistent over time [81]. Similarly, many studies suggest that shy individuals increase their mobile phone use to avoid face-to-face contact, which leads to PMPU [82,83,84].

Research concerning parenting styles suggests that parents should adopt adaptive parenting that is suitable to their child’s temperament [85]. Studies have found that inhibitive temperament (similar to shyness) mediates the relation between maternal authoritative parenting and girls’ prosocial behaviors [86]. Zarra-Nezhad et al. [87] discovered that parental emotional support positively predicted prosocial behavior only among shy children, while parental control positively predicted prosocial behaviors only among children who were not shy. Because parental mediation strategies are similar to parenting styles to some degree [88], parental monitoring of children’s mobile phone use should be similar to parental control. Thus, it could be inferred that parental monitoring exerts different effects based on the shyness level of a child. Furthermore, a study [89] showed that shyness also predicts an individual’s drinking motivation; shy individuals are more inclined to reduce their negative emotions and navigate social contexts via drinking, which can eventually lead to alcohol addiction. This suggests that shyness is related to behavioral motivation. Therefore, we hypothesized that parental monitoring would positively predict more escape motivation for shy adolescents than for those who are not shy.

### 1.4. The Present Study

Based on the above, the present study aimed to investigate the mechanism and impact of parental monitoring on adolescent PMPU through a hypothesized moderated mediation model (Figure 1). We anticipated that (a) parental monitoring would positively predict adolescent PMPU (H1) via the mediation of escape motivation (H2), and (b) adolescent level of shyness would moderate the impact of parental monitoring on adolescent escape motivation (H3) and PMPU (H4).

## 2. Materials and Methods

### 2.1. Participants

In the present study, we contacted an urban middle school in Beijing, China, and informed them of the purpose of our study. All students recruited in grades 7, 8, 10, and 11 were voluntary to participate. A total of 584 students were recruited. The mean age of these participants was 16.13 years (standard deviation (SD) = 2.80) with a range from 13 to 18; 267 (45.7%) were boys, 281 (49.0%) were girls, and 31 (5.3%) did not report gender. Among these students, 23.3% of fathers and 25.8% of mothers received a high-school education or below and 76.7% of fathers and 74.2% of mothers received an undergraduate level of education or above.

### 2.2. Procedures

We obtained the approval to conduct the study from the Research Ethics Committee of a major research university in Beijing and the principals of the participating schools. The students were informed of the voluntary nature of this study and their right to opt out at any time during the course of the study. Then, they were asked to complete a paper-pencil questionnaire that included demographic information, Parental Monitoring, Escape Motivation, Shyness and Problematic mobile phone-use measures.

### 2.3. Measures

#### 2.3.1. Parental Monitoring

Parental monitoring was assessed by the Parental Mediation of Children’s Internet Use Scale [18], which was validated in Chinese and exhibited satisfactory reliability and validity [90]. In this study, the word “internet use” was changed to “mobile phone use” to assess parental monitoring of children’s mobile phone-use behaviors. The scale included four items (e.g., “My parents check my mobile phone use behaviors; My parents check my mobile phone chatting records; My parents keep an eye on me when I use mobile phone”) on a 5-point Likert scale (1 = *never*, 5 = *always*). Higher scores reflected higher levels of parent monitoring of children’s mobile phone use. Cronbach’s α for the present study was 0.892. 

#### 2.3.2. Escape Motivation

Escape motivation was assessed by the Mobile Phone-Use Motivations Scale [49]. It consisted of six items (e.g., I use/play with my smartphone to feel less lonely; to fill uncomfortable silence; to make myself feel better when I feel down) using a 5-point scale ranging from 1 (*strongly disagree*) to 5 (*strongly agree*); higher scores indicated a higher degree of escape motivation. This measure yielded a Cronbach’s α of 0.865 in the present study.

#### 2.3.3. Shyness

Shyness was assessed by the Shyness Scale [72], which was validated in Chinese and showed satisfactory reliability and validity [91]. This scale included 13 items (e.g., I have trouble looking someone right in the eye; I feel tense when I’m with people I don’t know well; I feel inhibited in social situations.) on a 5-point Likert scale, ranging from 1 (*strongly disagree*) to 5 (*strongly agree*), with higher scores indicating a higher degree of shyness. Cronbach’s α for the scale was 0.803 in the present study.

#### 2.3.4. Problematic Mobile Phone Use (PMPU)

Problematic mobile phone use (PMPU) was assessed by the Problematic Mobile Phone-Use Scale [92], which was translated into Chinese and proved to be valid [93]. It consisted of 10 items (e.g., I find it difficult to switch off my mobile phone; I feel anxious if I have not checked for messages or switched on my mobile phone for some time; I find myself engaged on the mobile phone for longer periods of time than intended) on a 5-point scale that ranged from 1 (*strongly disagree*) to 5 (*strongly agree*), with higher scores indicating a higher degree of problematic mobile phone use. Cronbach’s α for the present study was 0.821.

#### 2.3.5. Statistical Analysis

We conducted descriptive analyses and Pearson correlations with SPSS 22.0 (IBM, New York, NY, USA). The pattern of missing data was first evaluated. The results showed that 1.25% of the data was missing, and the missing rates on all variables were less than 10%. Therefore, we used the listwise method to handle the missing data in the following structural equation model [94]. Among the 583 participants, 518 provided complete data on all the variables. Next, the moderated mediation model was tested using the SPSS macro PROCESS 3.0 (model 8) (http://www.afhayes.com) recommended by Hayes [95]. We generated 5000 bootstrapped samples to estimate the confidence interval of the model effect. A 95% confidence interval without zero indicates statistical significance.

## 3. Results

### 3.1. Preliminary Analyses

The descriptive statistics and correlation matrix are presented in Table 1. Parental monitoring was positively correlated with children’s escape motivation, shyness, and PMPU. Children’s escape motivation was positively correlated with their shyness level and PMPU. Children’s shyness level was positively correlated with PMPU.

### 3.2. Testing for the Proposed Model

The analysis results of SPSS macro PROCESS are presented in Table 2, which consists of four parts: Model 1, Model 2, conditional indirect effect analysis of Model 1, and conditional direct effect analysis of Model 2. Model 1 was used to test the effects of parental monitoring on children’s escape motivation (part of H2), and the interaction between parental monitoring and children’s shyness on children’s escape motivation (H4), after controlling for age and gender. Model 2 examined the effects of parental monitoring on children’s PMPU (H1), children’s escape motivation on children’s PMPU (part of H2), and the interaction between parental monitoring and children’s shyness on children’s PMPU (H3).

The conditional indirect effect analysis of Model 1 tested the effects of parental monitoring at its mean, plus one, and minus one standard deviation on children’s escape motivation at the mean of shyness. The conditional direct effect analysis of Model 2 tested the effects of parental monitoring at its mean, plus one standard deviation, and minus one standard deviation on children’s PMPU at the mean of the shyness. According to Model 1 (*F* = 5.96, R^2^ = 0.05, *p* < 0.001) and Model 2 (*F* = 65.74.45, R^2^ = 0.41, *p* < 0.001), after controlling for gender and age, parental monitoring positively predicted children’s PMPU (β = 0.078, *p* < 0.05), supporting H1 (see Figure 2). Parental monitoring positively predicted children’s escape motivation (β = 0.101, *p* < 0.05), and children’s escape motivation positively predicted children’s PMPU (β = 0.612, *p* < 0.001), supporting H2.

The interaction of parental monitoring and shyness showed significant effects on children’s escape motivation (β = 0.102, *p* < 0.05). Thus, H4 was supported. This finding suggests that the relation between parental monitoring and children’s escape motivation was moderated by children’s level of shyness (see Figure 3). In addition, two of the three conditional indirect effects (based on the moderator values at the mean and at plus and minus one standard deviation) were positive and significantly different from zero (see conditional indirect effect analysis of Model 1). That is, according to the interaction of parental monitoring and children’s shyness, the indirect effects of parental monitoring on children’s escape motivation were stronger when children’s shyness level was moderate to high, but lower when children’s shyness level was low.

Furthermore, the interaction of parental monitoring and shyness showed marginally significant effects on children’s PMPU (β = 0.063, *p* < 0.10). Therefore, H3 was supported. Two of the three conditional direct effects (based on the moderator values at the mean and at plus and minus one standard deviation) were positive and significantly different from zero (see conditional direct effect analysis of Model 2). These findings suggest that the relation between parental monitoring and children’s PMPU was moderated by children’s level of shyness (see Figure 4). The direct effects of parental monitoring on children’s PMPU were higher when children’s shyness was moderate to high, but lower when children’s shyness was low. In conclusion, the above results indicated that parental monitoring affects children’s PMPU through a moderated mediation path, with children’s escape motivation as the mediator and children’s shyness level as the moderator.

## 4. Discussion

To reduce the negative impact of children’s mobile phone use, parental monitoring has not achieved consistent results [45,48]; its underlying mechanism remains unclear. The present study examined a moderated mediation model and found that children’s escape motivation partially mediated the association of parental monitoring and their PMPU. In addition, children’s degree of shyness moderated the path from parental monitoring to their escape motivation and to PMPU.

### 4.1. Escape Motivation Partially Mediates the Relation between Parental Monitoring and Adolescent PMPU

Parental monitoring positively correlated with adolescent PMPU, which supports H1. This is consistent with the findings of previous studies regarding internet addiction [46]. Research has shown that adolescents are especially vulnerable to the negative impacts of increased mobile phone use, and parents hope to reduce these negative influences via mediation. Considering that teenage years are a critical period of learning and social development, most parents employ monitoring strategies to mediate their children’s mobile phone use. However, research findings suggest that parental monitoring often leads to unexpected results. Because of adolescents’ desire for freedom and psychological reactance induced by restriction, they may be subjected to the “Pandora effect” [63,68,96]; as the strength of parental monitoring of mobile phone use increases, adolescents’ mobile phone-use behaviors also increase, which can eventually lead to PMPU. Similar results were found in other areas of addiction, including adolescent internet addiction [97], internet dependency [98], and sexual behaviors [99]. From the perspective of the sociology of emotions, emotions serve as an important precursor to effective parental mediation. However, parental monitoring often results in the conflicts between parents and child, causing negative emotions. The model of compensatory internet use suggests that internet use is viewed as a compensatory means to escape the reality. An individual with relatively low overall happiness tends to relieve negative emotions and escape from real-life problems via mobile phone use [100,101,102]. Moreover, in the family environment, whether the parents’ supervision is effective or not partially depends on their own behaviors. According to Bandura’s social cognitive theory, children observe the behaviors of others around them and are especially prone to observing, and imitating their parents’ behaviors [103]. It has been found that parents’ looking down at their own mobile phone (parent phubbing) in the process of communication with their children will not only aggravate children’s addiction [104], but also affect children’s attitude towards self-control mobile phone use [105]. Therefore, if parents want to achieve efficient monitoring results, they should control their own mobile phone-use behaviors. In general, although parental monitoring aims to improve children’s mobile phone-use behaviors, it is often ineffective or leads to undesirable outcomes.

In addition, as we hypothesized, escape motivation mediated the relation between parental monitoring and adolescent PMPU, which supports H2. This result is consistent with similar studies. One study found that escape motivation mediated the predictive effect of negative emotions on PMPU [67]. When individuals face real-life struggles, escape motivation can prompt them to compensate through video games and/or the internet, which can eventually lead to video game or internet addiction [54,106,107]. Furthermore, Erikson [108] proposed that there is a developmental issue in every psychosocial stage of human development; adolescence is a crucial period for identity development, as well as a time when psychological reactance peaks. To adolescents, parental monitoring acts as a restriction of freedom, which could induce their escape motivation. In addition, parental monitoring often creates parent-child conflict, which could cause adolescents to develop negative emotions, thus leading to escape motivation.

### 4.2. Shyness Moderates Both the Relations of Parental Monitoring with Adolescent Escape Motivation and Adolescent PMPU

The impact of parental monitoring on adolescent PMPU differs across individuals. The present study found that adolescent shyness level moderated the relation between parental monitoring and adolescent escape motivation, and between parental monitoring and adolescent PMPU, which supports H4 and H3. As children’s shyness level increased, increases in parental monitoring strengthened their escape motivation and PMPU. This could be explained by the fact that shy individuals rely more on mobile phones to socialize, and parental monitoring can cause them to generate more negative reactions, such as psychological reactance and feelings of insecurity; this in turn perpetuates their escape motivation and exacerbates PMPU. There are three possible reasons why shy individuals rely more on mobile-phone socialization. First, shyness can be related to social anxiety; to avoid the embarrassment and discomfort elicited by face-to-face interactions, shy individuals tend to socialize via the internet, thus increasing internet use behaviors that can lead to internet and mobile-phone addiction [83,109]. Second, shy individuals often avoid occasions that expose them to evaluation [110]. Internet and mobile phone interactions can disguise their identity and thus protect them from others’ evaluations. Third, shyness that originates from social anxiety can expose individuals to psychological challenges [111]. To mitigate and overcome these challenges, shy individuals tend to avoid face-to-face interactions, devote themselves to internet use, and achieve satisfaction through online interactions. Overall, when their parents employ monitoring strategies, shy adolescents might feel insecure and avoid being evaluated, which prompts them to escape reality via mobile phone use.

### 4.3. Limitation and Implication

The present study has a few limitations. First, our data are cross-sectional, and thus cannot infer strong causational relationships. Experimental or longitudinal designs could be used to further prove the relations between these variables. Second, all of our data came from adolescents’ subjective responses. Although our measures have relatively high reliability/validity, the addition of responses from other sources (such as parents) would make our results more persuasive. Data of parental monitoring were obtained from children’s self-report, but not from their parents’ reports due to the limitation of our research conditions. In future research, obtaining data of parental monitoring from parents will help to reduce potential bias. Third, the findings in this study are in the context of Chinese culture, and its generalization to other culture should be made with caution. In order to avoid the negative effects of mobile phones, many parents in China prohibit children from bringing mobile phones to school. Using mobile phones at home is also often restricted. In comparison to children in other cultural contexts, Chinese children might have less freedom to use mobile phones. Moreover, studies on parental monitoring and children’s PMPU have yielded different results under different cultural and national contexts (such as Turkey and Germany) [19,47]. Therefore, future studies can explore the effects of cultural differences and cultural contexts on the effect of parental monitoring on children’s PMPU. Finally, future research should involve more schools and students to explore the differences between children in different developmental stages, such as early adolescence (12–13 years) and late adolescence (16–18).

Despite the above limitations, the contributions of the present study are relevant to educators, parents, and adolescents. First, our study results found that parental monitoring positively predicts adolescents’ PMPU. However, as the forms of online activities increase and the access to the internet becomes easier, parents are more likely to use monitoring strategies [20]. Moreover, parental monitoring is the most direct mediation method and, therefore, the most convenient for parents [18]. One study points out parents’ greater tendency to monitor their child’s mobile phone-use behaviors when their use increases [46]. When parental monitoring increases children’s mobile phone use, parental monitoring in turn increases, leading to a maladaptive cycle. Parents and educators should be cautious and optimize the use of monitoring strategies to control adolescent mobile phone use. Second, the present study found that parental monitoring positively predicted adolescents’ escape motivation, which predicted their PMPU. Third, the effect of parental monitoring on adolescents differed across temperaments. When parents strengthen their monitoring, shy adolescents are more likely to display escape motivation and develop problematic mobile phone-use behaviors. Parents should adopt appropriate mediation strategies according to their child’s temperament. 

## 5. Conclusions

Parental monitoring positively predicts children’s PMPU, and this predictive effect is partially mediated by children’s escape motivation. Children’s level of shyness moderates the relation between parental monitoring and children’s escape motivation, and the relation between parental monitoring and children’s PMPU, Thus, increasing the strength of parental monitoring can lead to an increase in escape motivation and PMPU among shy adolescents.

## Figures and Tables

**Figure 1 ijerph-17-01487-f001:**
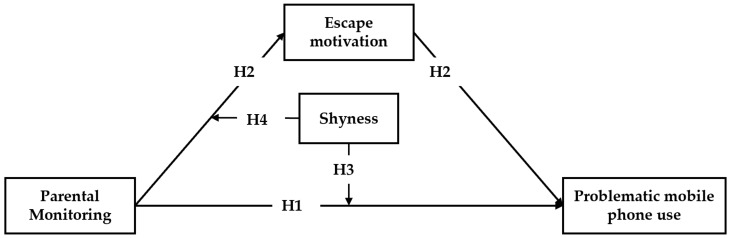
The hypothesized model.

**Figure 2 ijerph-17-01487-f002:**
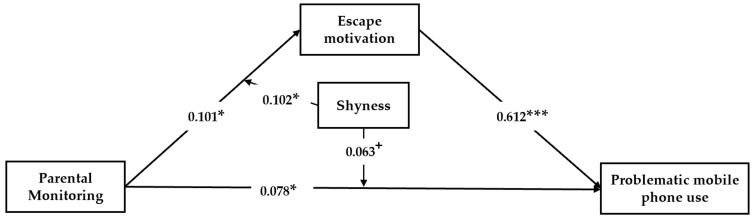
Moderation effects of shyness in the mediation model. All values shown are standardized coefficients. ^+^
*p* < 0.10. * *p* < 0.05. *** *p* < 0.001.

**Figure 3 ijerph-17-01487-f003:**
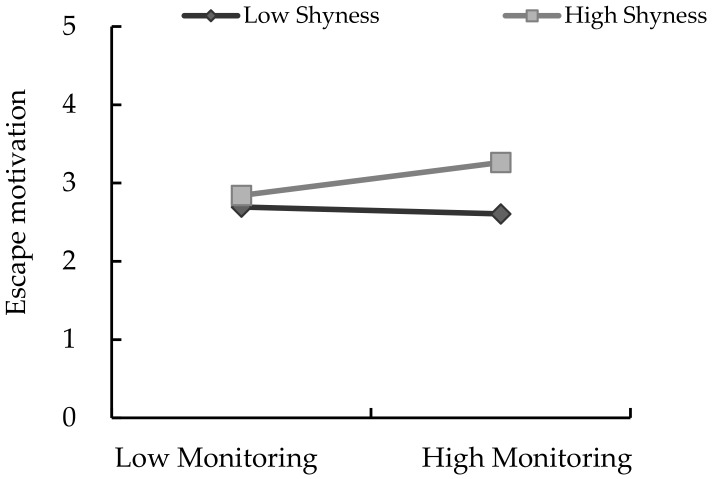
Moderation effect of shyness in the relation between parental monitoring and children’s escape motivation.

**Figure 4 ijerph-17-01487-f004:**
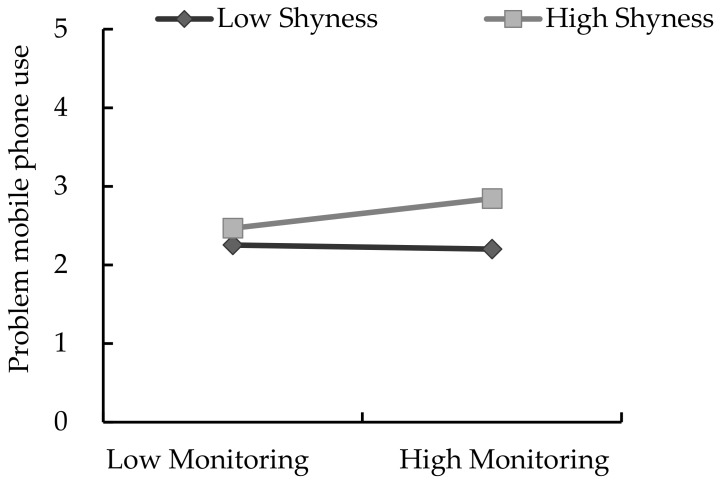
Moderation effect of shyness the relation between parental monitoring and children’s PMPU.

**Table 1 ijerph-17-01487-t001:** Means, standard deviations, and correlations between variables.

Variables	M	SD	1	2	3	4	5
1. Age	16.13	2.80	_				
2. Parental monitoring	2.06	1.14	0.114 **	_			
3. Escape motivation	2.86	1.05	−0.023	0.122 **	_		
4. Shyness	2.75	0.70	−0.031	0.135 **	0.147 ***	_	
5. PMPU	2.45	0.76	−0.092 *	0.163 ***	0.607 ***	0.213 ***	_

Note. PMPU = Problematic Mobile Phone Use. * *p* < 0.05. ** *p* < 0.01. *** *p* < 0.001.

**Table 2 ijerph-17-01487-t002:** Bootstrap test on moderated mediation effect.

Conditional Process Analysis	β	*SE*	*t*	*p*	*LLCI–ULCI*
Model 1					
Outcome: Escape motivation					
Predictors:					
Age	−0.009	0.014	−0.668	0.505	−0.037–0.018
Gender	0.150	0.079	1.902	0.058	−0.005–0.305
Parental monitoring	0.101 *	0.042	2.392	0.017	0.018–0.183
Shyness	0.129 **	0.041	3.164	0.002	0.050–0.209
Monitoring × Shyness	0.102 *	0.041	2.474	0.014	0.021–0.183
Model 2					
Outcome: PMPU					
Predictors:					
Age	−0.031 **	0.012	−2.672	0.008	−0.054–0.008
Gender	−0.156 *	0.066	−2.352	0.019	−0.287–0.026
Parental monitoring	0.078 *	0.035	2.198	0.028	0.008–0.148
Escape motivation	0.612 ***	0.035	17.571	0.000	0.544–0.681
Shyness	0.124 ***	0.035	3.582	0.000	0.056–0.191
Monitoring × Shyness	0.063 ^+^	0.035	1.806	0.071	−0.005–0.131
Conditional indirect effect analysis of model 1			β	*Boot SE*	*BootLLCI-BootULCI*
M − 1 SD			0.007	0.060	−0.111–0.125
M			0.103 *	0.042	0.021–0.186
M + 1 SD			0.188 ***	0.051	0.087–0.289
Conditional direct effect analysis of model 2			β	*Boot SE*	*BootLLCI-BootULCI*
M − 1 SD			0.020	0.050	−0.079–0.119
M			0.080 *	0.035	0.010–0.149
M + 1 SD			0.132 **	0.044	0.046–0.217

Note. Bootstrap sample size = 5000. SE = standard error, LL = low limit, CI = confidence interval, UL = upper limit. ^+^
*p* < 0.10. * *p* < 0.05. ** *p* < 0.01. *** *p* < 0.001.

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
