# Peer review of "Parental Monitoring and Adolescent Problematic Mobile Phone Use: The Mediating Role of Escape Motivation and the Moderating Role of Shyness"

_ijerph, 2020, doi:10.3390/ijerph17051487_

Round 1

Reviewer 1 Report

Introduction is too long, part of valuable information about role of escape motivation, moderating role of shyness can be used in the discussion section.

Please make shorter introduction part.

Materials and methods section: No information about permission of bioethics to perform this study. Also, lack of information, how students were selected, which part of all students they are representing? response rate? Do not described methods of statistical analysis. It can be placed at the separate section.

Presentation of results must be done in the conventional way. Strange UL, LL must be changed to limits like 0.021-0.186. It is not clear what is SE? Model 1 and Model 2 hypotheses must be described more detail. Table 2 name must be more detail.

Discussion section must include Chinese cultural context which you mention in the summary. Do results from you investigation can be applied in other countries, in other cultures?

Author Response

Response Letter

Comments from the Reviewer 1 and Responses:

Q1) Introduction is too long, part of valuable information about role of escape motivation, moderating role of shyness can be used in the discussion section.

Please make shorter introduction part.

Responses:

According to your suggestion, we have reorganized introduction part, especially the content about the role of escape motivation and shyness, to make it more concise (Please see the revised introduction). Besides, some valuable information in introduction has been reiterated in discussion section: “From the perspective of the sociology of emotions, emotions serve as an important precursor to effective parental mediation. However, parental monitoring often results in the conflicts between parents and child, causing negative emotions. The model of compensatory internet use suggests that internet use is viewed as a compensatory means to escape the reality. An individual with relatively low overall happiness tends to relieve negative emotions and escape from real life problems via mobile phone use [100–102].” “…Yet, as the forms of online activities increase and the access to the internet becomes easier, parents are more likely to use monitoring strategies [20].”.

Q2) Materials and methods section: No information about permission of bioethics to perform this study. Also, lack of information, how students were selected, which part of all students they are representing? response rate? Do not described methods of statistical analysis. It can be placed at the separate section.

Responses:

We have added “2.2 Procedures”, including permission of bioethics to perform this study; and “2.3.5 Statistical analysis”, including the response rate. In addition, in “2.1 Participants”, we added description about how students were selected.

Q3) Presentation of results must be done in the conventional way. Strange UL, LL must be changed to limits like 0.021-0.186. It is not clear what is SE? Model 1 and Model 2 hypotheses must be described more detail. Table 2 name must be more detail.

Responses:

According to your suggestion, we have made revision for LL-UL in Table 2, and added note for SE. We added detailed description about the test in Model 1, Model 2, and in the results of “3.2”. The title of Table 2 was changed to “Bootstrap test on Moderated mediation effect”.

Q4) Discussion section must include Chinese cultural context which you mention in the summary. Do results from you investigation can be applied in other countries, in other cultures?

Responses:

We added the following content in the discussion section:

Third, the findings in this study are on the basis of Chinese culture, and its generalization to other culture should be made with caution. In order to avoid the negative effects of mobile phones, many parents in China prohibit children from taking mobile phones to school. Using mobile phones at home is also often restricted. Perhaps compared with other cultural backgrounds, Chinese children have less freedom to use mobile phones...

Reviewer 2 Report

This is an interesting piece of work

Page 2, 1 para: there is some literature missing on mobile phone use in adolescents that should be added:

Randler, C., Wolfgang, L., Matt, K., Demirhan, E., Horzum, M. B., & BeÅŸoluk, Åž. (2016). Smartphone addiction proneness in relation to sleep and morningness–eveningness in German adolescents. Journal of behavioral addictions, 5(3), 465-473.

Demirhan, E., Randler, C., & Horzum, M. B. (2016). Is problematic mobile phone use explained by chronotype and personality?. Chronobiology international, 33(7), 821-831.

These studies use two different scales and one of the study in addition is also assessing personality.

Line 83: Bandura’s theory of learning from models: I am unsure if there is any literature on this, so I suggest the following: If you find some literature, please make a paragraph in the discussion, otherwise, please write on or two sentences in the discussion. I “feel” that parents monitoring is not very effective when parents themselves do not serve as some kind of role model. IN many cases, parents try to control their adolescents, but they (the parents) use the internet all the time, so that the children see their parents as role model. The best strategy would be to observe parents how they interact – there is some literature on parenting and mobile phone, but probably only concerning kindergarten stage and playground: while the children are playing, parents always use theis smartphones, which serves as a model. I think, this is one of the keys: interventions should not focus only on the poor adolescents, but should also take adults into account.

Line 111: perfectly, reactance is a main aspect.

Line 159: probably write “showed” ?

Line 189: Cronbach for the present study or for the original study, please specify

Line 200: similar question: alpha for present sample?

Line 207: similar question: alpha for present sample?

Concerning all the translation/validity checks for the Chinese scales: I would be happy to see a little bit more information, i.e., one or two sentences about the scales additionally (not much more), but it is vague just to hear they are valid.

Please include a sections Statistical analyses, there you can report which macros you used. Where there any missing data, how were missing data treated?

Was it an online survey? Was it paper pencil?

Figures: do the figures show really data that are related (with lines)? Otherwise I suggest using other charts. I know that SPSS makes these graphs, but they are conceptually wrong. IN the SPSS26 version you can changes them to bar graphs, perhaps also in the version 25.

Line 342: this is a strong argument. How do you define temperament? In terms of personality of Cloninger’s TCI? You should either elaborate further on this topic or probably delete the sentences.

Author Response

Comments from the Reviewer 2 and Responses:

This is an interesting piece of work

Q1) Page 2, 1 para: there is some literature missing on mobile phone use in adolescents that should be added:

Randler, C., Wolfgang, L., Matt, K., Demirhan, E., Horzum, M. B., & BeÅŸoluk, Åž. (2016). Smartphone addiction proneness in relation to sleep and morningness–eveningness in German adolescents. Journal of behavioral addictions, 5(3), 465-473.

Demirhan, E., Randler, C., & Horzum, M. B. (2016). Is problematic mobile phone use explained by chronotype and personality?. Chronobiology international, 33(7), 821-831.

These studies use two different scales and one of the study in addition is also assessing personality.

Responses:

According to your suggestion, we have added the references in the corresponding positions.

Q2) Line 83: Bandura’s theory of learning from models: I am unsure if there is any literature on this, so I suggest the following: If you find some literature, please make a paragraph in the discussion, otherwise, please write on or two sentences in the discussion. I “feel” that parents monitoring is not very effective when parents themselves do not serve as some kind of role model. IN many cases, parents try to control their adolescents, but they (the parents) use the internet all the time, so that the children see their parents as role model. The best strategy would be to observe parents how they interact – there is some literature on parenting and mobile phone, but probably only concerning kindergarten stage and playground: while the children are playing, parents always use theis smartphones, which serves as a model. I think, this is one of the keys: interventions should not focus only on the poor adolescents, but should also take adults into account.

Responses:

According to your suggestion, we added “Moreover, in the family environment, whether the parents’ supervision is effective or not partially depends on their own behaviors. According to Bandura’s social cognitive theory, children observe the behaviors of others around them and are especially prone to observing, and imitating their parents’ behaviors [103]. It has been found that parents’ looking down at their own mobile phone (parent phubbing) in the process of communication with their children will not only aggravate children’s addiction [104], but also affect children’s attitude towards self-control mobile phone use [105]. Therefore, if parents want to achieve efficient monitoring results, they should control their own mobile phone use behaviors.” in Discussion section.

Q3) Line 111: perfectly, reactance is a main aspect.

Responses:

Thank you for your kind note.

Q4) Line 159: probably write “showed” ?

Responses:

Yes, we have revised it to “showed.”

Q5) Line 189: Cronbach for the present study or for the original study, please specify

Responses:

It’s for the present study, and we added related description.

Q6) Line 200: similar question: alpha for present sample?

Responses:

Yes, it is for the present sample, and we added description now.

Q7) Line 207: similar question: alpha for present sample?

Responses:

Yes, it is for the present sample, and we added description now.

Q8) Concerning all the translation/validity checks for the Chinese scales: I would be happy to see a little bit more information, i.e., one or two sentences about the scales additionally (not much more), but it is vague just to hear they are valid.

Responses:

According to your suggestion, we have added 2 sample items to each scale in the Materials and methods section as follows.

Parental Monitoring:

  1. My parents check my mobile phone use behaviors;
  2. My parents check my mobile phone chatting records;
  3. My parents will keep an eye on me when I use mobile phone.

Escape Motivation:

I use / play with my smartphone

  1. To feel less lonely;
  2. To fill uncomfortable silence;
  3. To make myself feel better when I feel down.

Shyness:

  1. I have trouble looking someone right in the eye;
  2. I feel tense when I'm with people I don't know well;
  3. I feel inhibited in social situations.

Problematic Mobile Phone Use:

  1. I find it difficult to switch off my mobile phone;
  2. I feel anxious if I have not checked for messages or switched on my mobile phone for some time;
  3. I find myself engaged on the mobile phone for longer periods of time than intended.

Q9) Please include a sections Statistical analyses, there you can report which macros you used. Where there any missing data, how were missing data treated?

Responses:

We have added “2.3.5 Statistical analysis,” including how missing data were treated. “We conducted descriptive analyses and Pearson correlations with SPSS 22.0. The pattern of missing data was first evaluated. The results showed that 1.25% of the data was missing, and the missing rates on all variables were less than 10%. Therefore, we used the listwise method to handle the missing data in the following structural equation model [94]. Among the 583 participants, 518 provided complete data on all the variables……”

Q10) Was it an online survey? Was it paper pencil?

Responses:

It is a paper pencil survey, and we have added the information in “2.2 Procedures”

“…… Then, they were asked to complete a paper-pencil questionnaire that included demographic information, Parental Monitoring, Escape Motivation, Shyness and Problematic mobile phone use measures.”

Q11) Figures: do the figures show really data that are related (with lines)? Otherwise I suggest using other charts. I know that SPSS makes these graphs, but they are conceptually wrong. IN the SPSS26 version you can changes them to bar graphs, perhaps also in the version 25.

Responses:

Yes, the data used in the diagram are the real data of the model.

Q12) Line 342: this is a strong argument. How do you define temperament? In terms of personality of Cloninger’s TCI? You should either elaborate further on this topic or probably delete the sentences.

Responses:

Thank you for your advice, we have deleted the sentences.

Reviewer 3 Report

Dear Authors I wanted to assess positively the study topic and its importance for prevention/intervention strategies. Online parental mediation is a very important construct and requires further research. I think it is a manuscript of interest and I value it positively, but the authors need to clarify and improve different parts of the manuscript: 1. The title should not indicate Parenting Monitor online or similar? It is about an online context. 1. The introduction is extraordinarily complete. I congratulate them, but I also think that it is excessive for presenting the hypotheses and could be restructured to be more concise and clear. 2. Although there are 4 clear hypotheses, I do not remember that there were research objectives. It is usual and necessary to have both in a scientific text. 3. There are parts of the "Materials and Methods" that require further clarification: a. There is no indication of approval by the appropriate ethics committee b. The authorization (consent) used is not indicated. c. In general, there should be a procedural section explaining how, where, who has been involved in the collection process. If the questionnaires have been collected on paper or online, the duration of the process, what they did with the questions, etc. d. There is no indication of whether it is a rural/urban setting or other details of the sample. e. I think the results should not have the explanation of the statistical analyses. There should be a section in the method (Statistical analysis) on this point that states what is to be carried out, the programme used, etc. These points should be removed from the results. f. The type of study design is not indicated. Results: 1. I think that you could better address the reader through the hypotheses that have been established previously. 2. I wonder if the variable sex and age might not be playing an important role in the results. What do the authors think? Have they analyzed the model for boys and girls? They should also look at whether there are differences between those children in early adolescence (12-13 years) and late adolescence (16-18) 3. Remove the variable "gender" from table 1. At the level of discussion (as an introduction and method) there is a key aspect: Parenting monitor is NOT evaluated in the parents, but the perception of the children about their behaviours (or intention of their behaviours). I think this point is key and requires further analysis in the introduction and discussion. Likewise, it is a limitation not to have used hetero reports with the parents and it is a line for the future. Best regards,

Author Response

Comments from the Reviewer 3 and Responses:

Dear Authors I wanted to assess positively the study topic and its importance for prevention/intervention strategies. Online parental mediation is a very important construct and requires further research. I think it is a manuscript of interest and I value it positively, but the authors need to clarify and improve different parts of the manuscript: 1.

Q1) The title should not indicate Parenting Monitor online or similar? It is about an online context.

Responses:

Thank you for your kind reminder. Yes, parental monitoring in this research is in an online context, involving parents’ monitoring in children’s mobile phone use, but not in learning or other aspects. We used “Parental monitoring” in the title, but not “Parental monitoring online” because we considered that the latter might lead to ambiguity (e.g., parents monitor children by using cyber-based approach).

Q2) The introduction is extraordinarily complete. I congratulate them, but I also think that it is excessive for presenting the hypotheses and could be restructured to be more concise and clear.

Responses:

According to your suggestion, we deleted the description of hypothesis in the introduction “1.1”, “1.2,” and “1.3,” and added specific hypothesis in “1.4”.

Q3) Although there are 4 clear hypotheses, I do not remember that there were research objectives. It is usual and necessary to have both in a scientific text.

Responses:

Thank you for your suggestion. In order to clearly express the purpose of the research, we added “Therefore, the current study aimed to further explore the impact of parental monitoring on children’s problematic mobile phone use and to investigate the mechanism underlying such an impact in the Chinese cultural context.”

There are parts of the "Materials and Methods" that require further clarification:

Q4) There is no indication of approval by the appropriate ethics committee

Responses:

We have added description in the “2.2 Procedures.” “We obtained the approval to conduct the study from the Research Ethics Committee of a major research university in Beijing and the principals of the participating schools…”

Q5) The authorization (consent) used is not indicated.

Responses:

We have added description in the “2.2 Procedures.” “…The students were informed of the voluntary nature of this study and their right to opt out at any time during the course of the study…”

Q6) In general, there should be a procedural section explaining how, where, who has been involved in the collection process. If the questionnaires have been collected on paper or online, the duration of the process, what they did with the questions, etc.

Responses:

We have added description in the “2.2 Procedures.” “We obtained the approval to conduct the study from the Research Ethics Committee of a major research university in Beijing and the principals of the participating schools. The students were informed of the voluntary nature of this study and their right to opt out at any time during the course of the study. Then, they were asked to complete a paper-pencil questionnaire that included demographic information, Parental Monitoring, Escape Motivation, Shyness and Problematic mobile phone use measures.”

Q7) There is no indication of whether it is a rural/urban setting or other details of the sample.

Responses:

It is in an urban setting, and We have added description in the “2.2 Procedures.” “In the present study, we contacted an urban middle school in Beijing, China, and informed them of the purpose of our study. All students recruited in grades 7, 8, 10, and 11 were voluntary to participate.”

Q8) I think the results should not have the explanation of the statistical analyses. There should be a section in the method (Statistical analysis) on this point that states what is to be carried out, the programme used, etc. These points should be removed from the results.

Responses:

We have added a section named “2.3.5 Statistical analysis”. “We conducted descriptive analyses and Pearson correlations with SPSS 22.0. The pattern of missing data was first evaluated. The results showed that 1.25% of the data was missing, and the missing rates on all variables were less than 10%. Therefore, we used the listwise method to handle the missing data in the following structural equation model [94]. Among the 583 participants, 518 provided complete data on all the variables. Next, the moderated mediation model was tested using the SPSS macro PROCESS 3.0 (model 8) (http://www.afhayes.com) recommended by Hayes [95]. We generated 5000 bootstrapped samples to estimate the confidence interval of the model effect. A 95% confidence interval without zero indicates statistical significance.”

Q9) The type of study design is not indicated.

Responses:

We have added description in “2.3.5 Statistical analysis.” “……Next, the moderated mediation model was tested using the SPSS macro PROCESS 3.0 (model 8) (http://www.afhayes.com) recommended by Hayes [95]. We generated 5000 bootstrapped samples to estimate the confidence interval of the model effect. A 95% confidence interval without zero indicates statistical significance.”

Results:

Q10) I think that you could better address the reader through the hypotheses that have been established previously.

Responses:

We added corresponding assumptions in the results section. “…parental monitoring positively predicted children’s PMPU (β = 0.078, p < 0.05), supporting H1 (see Figure 2). Parental monitoring positively predicted children’s escape motivation (β = 0.101, p < 0.05), and children’s escape motivation positively predicted children’s PMPU (β = 0.612, p < 0.001), supporting H2.”

Q11) I wonder if the variable sex and age might not be playing an important role in the results. What do the authors think? Have they analyzed the model for boys and girls? They should also look at whether there are differences between those children in early adolescence (12-13 years) and late adolescence (16-18)

Responses:

Thank you for your inspirable suggestions. In fact, gender showed non-significant correlation with all the other variables, so we treated gender as a control factor in the further analysis.

It is also an interesting research point to address the differences between children in early adolescence (12-13 years) and late adolescence (16-18), however, we did not compare the differences because the number of corresponding subjects in this study is not sufficient and may cause data deviation. We will consider involving more schools and students in the future, and further explore the differences between children in different age. We have addressed this in the Limitation section.

Q12) Remove the variable "gender" from table 1.

Responses:

According to your suggestion, we removed gender.

Q13) At the level of discussion (as an introduction and method) there is a key aspect: Parenting monitor is NOT evaluated in the parents, but the perception of the children about their behaviours (or intention of their behaviours). I think this point is key and requires further analysis in the introduction and discussion. Likewise, it is a limitation not to have used hetero reports with the parents and it is a line for the future.

Responses:

Thank you for your advice. Actually, in the parental monitoring scale, we asked children to objectively evaluate their parents’ monitoring behaviors (e.g., My parents check my mobile phone use behaviors; My parents check my mobile phone chatting records, etc.), rather than perceived feelings, to lower the bias between children and parents. It’s true that it would be more convincing if parents answered the parental monitoring scale, but we did not collect the data of parents due to the limitation of research conditions, and we will try our best to overcome this difficulty in the future research.

We also added description in the limitations “…Data of parental monitoring were obtained from children’s self-report, but not from their parents’ reports due to the limitation of our research conditions. In the future research, obtaining data of parental monitoring from parents will help to reduce potential bias…”

Round 2

Reviewer 3 Report

Thank you for your detailed response to all points.
If the editor considers your answers sufficient, I would recommend their publication.
Best regards,